# Association between ApoE ε4 Carrier Status and Cardiovascular Risk Factors on Mild Cognitive Impairment among Mexican Older Adults

**DOI:** 10.3390/brainsci11010068

**Published:** 2021-01-07

**Authors:** Sara G. Aguilar-Navarro, Itzel I. Gonzalez-Aparicio, José Alberto Avila-Funes, Teresa Juárez-Cedillo, Teresa Tusié-Luna, Alberto Jose Mimenza-Alvarado

**Affiliations:** 1Department of Geriatric Medicine, Instituto Nacional de Ciencias Médicas y Nutrición Salvador Zubirán, Mexico City 14080, Mexico; sgan30@hotmail.com (S.G.A.-N.); avilafunes@live.com.mx (J.A.A.-F.); 2Geriatric Medicine & Neurology Fellowship, Instituto Nacional de Ciencias Médicas y Nutrición Salvador Zubirán, Mexico City 14080, Mexico; slidr_irais@hotmail.com; 3Bordeaux Population Health Research Center, INSERM-University of Bordeaux, UMR 1219, F-33000 Bordeaux, France; 4Epidemiologic and Health Service Research Unit, Aging Area, Instituto Mexicano del Seguro Social, Centro Médico Nacional Siglo XXI, Mexico City 06720, Mexico; terezillo@yahoo.com.mx; 5Unidad de Biología Molecular y Medicina Genómica, Instituto Nacional de Ciencias Médicas y Nutrición Salvador Zubirán e Instituto de Investigaciones Biomédicas, UNAM, Mexico City 14080, Mexico; mttusie@gmail.com

**Keywords:** apolipoprotein E ε4, cardiovascular risk factors, amnestic MCI, non-amnestic MCI, Mexican Mestizo older adults

## Abstract

Mild cognitive impairment (MCI) (amnestic or non-amnestic) has different clinical and neuropsychological characteristics, and its evolution is heterogeneous. Cardiovascular risk factors (CVRF), such as hypertension, diabetes, or dyslipidemia, and the presence of the Apolipoprotein E ε4 (ApoE ε4) polymorphism have been associated with an increased risk of developing Alzheimer’s disease (AD) and other dementias but the relationship is inconsistent worldwide. We aimed to establish the association between the ApoE ε4 carrier status and CVRF on MCI subtypes (amnestic and non-amnestic) in Mexican older adults. Cross-sectional study including 137 older adults (*n* = 63 with normal cognition (NC), *n* = 24 with amnesic, and *n* = 50 with non-amnesic MCI). Multinomial logistic regression models were performed in order to determine the association between ApoE ε4 polymorphism carrier and CVRF on amnestic and non-amnestic-MCI. ApoE ε4 carrier status was present in 28.8% participants. The models showed that ApoE ε4 carrier status was not associated neither aMCI nor naMCI condition. The interaction term ApoE ε4 × CVRF was not statistically significant for both types of MCI. However, CVRF were associated with both types of MCI and the association remained statistically significant after adjustment by sex, age, and education level. The carrier status of the ApoE genotype does not contribute to this risk.

## 1. Introduction

The spectrum of cognitive impairment in older adults ranges from normal cognitive status, cognitive complaint with normal screening test, mild cognitive impairment (MCI) to dementia [1].

MCI, refers to the intermediate stage between the expected cognitive decline, associated with normal aging and dementia, which has gained a lot of interest. MCI represents a damage of cognitive skills, revealed by neuropsychological tests—with global cognitive functions and preserved everyday activities [2], which confers a higher risk of conversion to dementia, particularly the amnestic MCI subtype Alzheimer´s dementia [3,4,5]. The MCI prevalence among older adults ranges from 3.1% to 37.5% according to the criteria used [6], which has made it difficult to establish its real frequency. In general, there are two subtypes of MCI: amnestic (aMCI), and non-amnestic (naMCI), each with specific neuropsychological characteristics, and which in turn, are subdivided into single- and multi-domain subtypes [7].

On the other hand, apolipoprotein E ε 4 (ApoE ɛ4) carrier status homozygous and heterozygous is the most important genetic risk factor both AD and MCI, while the other two isoforms (ε2 and ε3) of the gene are protective [8,9]. However, the association between ApoE ε4 and the risk of AD does not seem to be universal. Population-based studies have shown a weak association between the ApoE ε4 polymorphism and AD among African-Americans, (ε4/ε4, OR: 5.7, 95% CI: 2.3–14.1) and Hispanic (ε4/ε4, OR: 2.2, 95% CI: 0.7–6.7) populations, compared with Japanese (ε4/ε4, OR: 33.1, 95% CI: 13.6–80.5) and non-Hispanic persons (ε4/ε4, OR: 12.5, 95% CI: 8.8–17.7) [10]. The presence of ApoE ε4 polymorphism combines synergistically with atherosclerosis, peripheral vascular disease or diabetes, which all contribute to an increased risk of cognitive decline, particularly AD [10]. In the same vein, ε4 allele has also been associated with increased susceptibility to vascular dementia, cerebral amyloid angiopathy (CAA), and cognitive decline during normal aging. In this way, ApoE ε4 is a risk factor for cardiovascular disease suggesting an interaction with cerebrovascular disease, which in turn, has a harmful effect on the cognitive function [9].

However, the association between ApoE ε4 polymorphism and the presence of cognitive impairment has been inconsistent in various populations around the world and whether the cause-effect association of the presence of CVRF and MCI is modulated by the ApoE ε4 polymorphism is also unclear. Therefore, the objective of this study was to establish the association between the ApoE ε4 carrier status and CVRF on MCI subtypes (amnestic and non-amnestic), as well as to determine if there is a modification of the effect due to the presence of CVRF and Apo E ε4 carrier status among Mexican older adults.

## 2. Materials and Methods

Cross-sectional study conducted at a Memory Clinic in a tertiary level University Hospital in Mexico City, between March 2018 and February 2020. All procedures were carried out only after written informed consent was obtained from the participants. The study was approved by the Institutional Ethics Research Committee (GER-2416-17-19).

### 2.1. Participants

Eligible participants were adults aged 60 or older, ambulatory, as well as mestizo-Mexicans. They underwent a comprehensive geriatric assessment by trained staff using standardized methods, and a neuropsychological evaluation was conducted by an expert neuropsychologist. Exclusion criteria included uncontrolled or untreated depressive symptoms ≥ 6 of the score of 15-items (Geriatric Depression Scale) (GDS) [11], delirium or previous diagnosis of dementia, visual or hearing impairment, illiterates, uncontrolled hypertension, untreated thyroid disease, high blood levels of glycated hemoglobin (≥9%), presence of severe heart failure, and recent traumatic brain injury. Magnetic resonance imaging was obtained for all participants using a standard protocol. Images were obtained with a 1.5 T resonator (Siemens^®^ Medical Systems, Waukesha, Wisconsin, USA), including whole-brain T1-weighted, T2-weighted, FLAIR, and T2*-weighted gradient-recalled echo. The MRI results were used by the researchers to rule out structural pathology that excluded the patient from the study and to classify MCI subtypes. 

Participants with aMCI, and naMCI subtype, and normal cognition (NC) were included. Diagnosis was established according to Petersen’s criteria [12,13]. The Petersen criteria include subjective memory complaint, corroborated by an informant, together with preserved everyday activities, a memory impairment based on a standard neuropsychological test, preserved global cognitive functions, and finally the exclusion of dementia [13].

The neuropsychologist evaluated the cognitive profiles through the following neuropsychological battery: Mini-Mental State Examination (MMSE) [14], Montreal Cognitive Assessment (MoCA) [15] and NEUROPSI (Brief Neuropsychological Evaluation in Spanish), a standardized neuropsychological test for the Mexican population [16]. This test succinctly assesses a broad spectrum of cognitive domains, including orientation, attention and concentration, language, memory, visuospatial skills, and executive functions. Normative data for grading the test stem, from validation performed in Mexicans, and is adjusted according to age, and educational level. The NEUROPSI has shown an appropriate test re-test reliability as well as substantial interrater agreement. A composite score of 1.5 standard deviations (SD) below the adjusted mean for age and education was considered as MCI. According to the evaluation, due to the sample size, it was decided to classify MCI into two predominant profiles: aMCI: (1) amnestic (memory impairment only and amnestic multidomain), or naMCI: (2) non-amnestic (single nonmemory cognitive domain and multidomain impaired). The applied criteria are based on the recommendations of the National Institute on Aging Alzheimer’s Association workgroups on diagnostic guidelines for Alzheimer’s disease [17].

The Katz index for activities of daily living (ADL) [18] and the Lawton Index for instrumental activities of daily living (IADL) were used for the assessment of functional status [19]. A participant was considered dependent for ADL when the score was ≤5/6; whereas a subject was considered dependent for IADL when the Lawton’ score was ≤7/8 for women and ≤4/5 for men, respectively.

The differentiation between MCI and NC was considered if participants denied having a memory complaint and had a normal cognitive performance to the battery of standardized neuropsychological tests, according to age, sex, and educational level.

### 2.2. ApoE Genotype Determination Method

The ApoE genotype was determined with the use of a polymerase chain reaction (PCR). 10 mL of peripheral blood was obtained, subsequently, and following the real time PCR usual method, DNA was extracted from leukocytes and later was amplified using oligonucleotide F4 (5′-ACAGAATTCGCCCCGGCCTGGTAcACAC-3′) and F6 (5′-lAAGCITGGCACGGCTGn = cAAG). Each reaction mixture was heated at 95 °C for 5 min for denaturation and subjected to 30 cycles for amplification, obtaining approximately 300 ng of amplified ApoE sequences. Subsequently, 5 units of Hal (New England Biolabs, Ipswich, MA, USA) were added for the digestion of the ApoE sequences. Each result was combined with polyacrylamide to perform electrophoresis, obtaining the different genotypes: ε2 ε2/ε2 ε3/ε3 ε3/ε3 ε4/ε4 ε4/ε4 ε2 [18]. For the present study, the genotype was operationalized as a binomial variable: those with 1 or more ApoE ε4 allele versus no ApoE ε4 allele.

### 2.3. Exposure Variable

Regarding the clinical assessment, in the clinical evaluation, some questions were asked about the presence and/or absence of the following conditions, related to the cardiovascular risks, which included: hypertension, current smoking status, obesity, hypothyroidism, dyslipidemia, diabetes, coronary heart disease (CHD), and history of stroke. Furthermore, the eight cardiovascular comorbidities were added to construct a variable for cardiovascular risk factors CVRF

### 2.4. Covariates

Socio-demographic variables included sex, age and educational level as continuous (years of age and education). 

### 2.5. Statistical Analysis

To calculate the minimum sample size of people over the age of 60 years necessary in this study to provide a valid estimate of the prevalence of MCI of 9%, a type I error of 0.05 and a power of 80% was assumed. Estimating that at least 36 patients per group were necessary to find the statistical power.

Variables were described using arithmetic means and standard deviations (SD) or frequency, and proportion, when appropriate. We used chi-square tests to compare categorical data and analysis of variance (ANOVA) for continuous variable. Post hoc Bonferroni analysis were used to identify intra-group differences. In addition, to determine the association, on the one hand, between the ApoE ε4 carrier status and, on the other, of the CVRF with aMCI, naMCI, multinomial logistic regression models were constructed, adjusting for potential confounders, including age, sex, and years of education. The interaction between ApoE ε4 and CVRF was analyzed to test the modifying effect on the dependent variable. Odds Ratio (OR) were estimated. A *p*-value < 0.05 was considered statistically significant, and 95% confidence interval (CI) were provided. All statistical analyses were performed using a SPSS version 22 for Windows^®^ (Chicago, IL, USA). 

## 3. Results

A total of 137 participants were allocated in three different subgroups: NC (*n* = 63), aMCI (*n* = 24), and naMCI (*n* = 50). The mean age was 71 years (±7.2), 72.3% were female and the mean educational years were 12.7 years (±5.04). Mean of MMSE was 27.8 (±1.9) while, 24.1 (±3.6) and 101.5 (±13.9) for the MoCA and Neuropsi test, respectively. Hypertension was the most frequent CVRF (57%), followed by dyslipidemia (39.4%), and diabetes (25.8%). The mean of CVRF was 1.9 (±1.4). Apo E carrier status were ε2/ε2 homozygous none, ε2/ε3 2 (0.7%), and ε3/ε3 homozygous 98 (71.8%), whereas ε3/ε4 heterozygous 36 (26.2%) and ε4/ε4 homozygous 3 (2.1%).

Table 1 presents the comparative analysis of sociodemographic and health-related characteristics according to neurocognitive status. As expected, the NC group was younger (*p* < 0.001), had more years of education (*p* < 0.001) and had better cognitive performance. There were statistically significant differences between the aMCI and naMCI in depressive symptoms by GDS scale (3.6 vs. 2.5 *p* < 0.005). The CVRF mean were higher in the aMCI 2.5 (±1.5) between NC and aMCI and naMCI group (*p* < 0.004).

Table 2 shows the univariate multinomial logistic regression due to the effect of ApoE ε4 carrier status in the three groups. Compared to the NC group, those with aMCI or naMCI were older and had more CVRF. Education has an inverse association both aMCI (*p* < 0.002), naMCI (*p* < 0.001) in comparison with the NC. Depressive symptoms were only associated with aMCI subgroup (*p* < 0.002). Regarding cognitive performance compared to NC, both aMCI had lower NEUROPSI test scores globally and in the domains of concentration and attention, memory, language, and executive functions.

The multinomial logistic regression model of cognitive function is showed in the Table 3. The unadjusted model did not show a statistically significant association between being ApoE ε4 carrier status and the presence of any type of MCI. On the contrary, there was a statistically significant association between the presence of CVRF and aMCI (*p* = 0.004) as well as naMCI (*p* = 0.009). This association remained statistically significant even after adjusting for age, sex, and education. Finally, the interaction term ApoE ε4 and CVRF was not statistically significant for both types of MCI.

## 4. Discussion

MCI amnestic or non-amnestic has different clinical and neuropsychological characteristics, and its trajectory is heterogeneous. Our study shows that the presence of factors, such as older age, low education, cardiovascular risk factors, and depression, are associated with aMCI and naMCI independently of the APOE e4 allele carrier status.

Genis et al., in a study that included 297 Mexicans with an unidentified cognitive disorder demonstrated that the ApoE ε4 genotype increased the risk of cognitive impairment by approximately 6% [20]. Juárez-Cedillo et al. demonstrated, in a study in a Mexican population that aimed to assess the prevalence of MCI in the Mexican population and that included 2944 subjects, that advanced age was strongly associated with the risk of MCI; with a (OR, 2.09; 95% CI, 1.44–2.94, *p* < 0.001) for the group between 71–84 years, and (OR, 5.13; 95% CI, 3.49–7.61, *p* < 0.001) in those older than 85 years. [5] Other study conducted among 175 Mexican older adults that evaluated the associated factors with the progression of MCI to dementia, showed older age as a risk factor to dementia (HR, 4.95; 95% CI, 1.96–12.46; *p* < 0.001) [21].

Another risk factor for developing MCI is low educational level. A study that evaluated the risk factors in Mexican-Americans to develop cognitive impairment, showed that the protective effect of the educational level offers a reduction in the prevalence only that those with more than 13 years of schooling, while lower levels increased the risk [22]. The effects of the years of education are based on the concepts of cognitive and brain reserve, in addition to the maintenance of cognitive reserve throughout life [23].

Among the cardiovascular factors previously linked to dementia and MCI, diabetes is the most associated risk factor for cognitive impairment in the Mexican population. Luchsinger et al. demonstrated, in a longitudinal cohort study that included 1772 participants, that diabetes was significantly associated with an increased risk, compared to all the different types of MCI (HR, 1.4; 95% CI, 1.1–1.8) after adjusting diverse variables, such as: age, sex, ethnic group, schooling level, APOEε4, hypertension, low-density lipoprotein level, heart disease, cerebrovascular disease, and smoking [24]. The Singapore Longitudinal Aging Study Cohort also demonstrated an association between older age an increase in the incidence of MCI (HR, 2.84; 95% CI, 1.92–4.19) [25].

Another of the linked factors has been hypertension: a systematic review, which included fourteen studies (*n* = 96,158) and which evaluated the association between the decrease in blood pressure and the cognitive prognosis during 49.2 months, showed that those patients with antihypertensive treatment had a reduction in the risk from dementia or MCI (12 trials; 92,135 participants) (7.0% vs. 7.5% of patients over a mean trial follow-up of 4.1 years; (OR, 0.93; 95% CI, 0.88–0.98); absolute risk reduction (OR, 0.3; 95% CI, 0.09–0.68), and cognitive decline (8 trials) (20.2% vs. 21.1% of participants over a mean trial follow-up of 4.1 years (OR, 0.93; 95 % CI, 0.88–0.99); absolute risk reduction, (0.71%, 95% CI, 0.19–1.2%) concluding that the decrease in blood pressure in the group with antihypertensive treatment was associated with a lower risk of incidence of dementia or cognitive impairment [26]. Similarly, the SPRINT-MIND study demonstrated that intensive blood pressure treatment (<120 mmHg systolic) reduced the risk of MCI (14.6 vs. 18.3 × 1000 people/year, (HR, 0.81; 95% CI, 0.69–0.95), emphasizing the importance of Blood Pressure control at this stage [27]. The mechanism by which hypertension produces alteration in cognitive function is through inducing atherosclerosis, remodeling, hypertrophy of large arteries, presence of microateromas and lipohyalinosis in small vessels, which leads to vascular damage, chronic hypoxia, and brain atrophy [28].

Regarding dyslipidemia, some studies evaluated the effect of this disease and the risk of developing MCI. Hashem HA et al., demonstrated in a study that included 158 adults over 65 years of age, that dyslipidemia (hypertriglyceridemia) was associated with the risk of developing MCI, compared to patients with lower levels (*p* < 0.001) [29]. Tze Ping et al. demonstrated in the Singapore Longitudinal Aging Study Cohort, which included 1519 participants, that dyslipidemia was associated with an increase in the incidence of MCI (HR, 1.48; 95% CI, 1.01–2.15) [25].

The mechanisms by which dyslipidemia favors neurodegeneration and vascular damage, seem to initiate extracellular accumulation of amyloid beta protein, alteration of synaptic connections, as well as increased oxidative stress in the hippocampus [30,31].

On the other hand, the association between depression and cognitive impairment is overwhelming. A meta-analysis study which included (*n* = 20,892) and whose objective was to know the prevalence of depression in patients with MCI, reported an overall prevalence of 32% in patients with MCI (95% CI, 27–37%) and 40% (95% CI, 32–48%), in clinical trials, showing that depression in this group of patients is high [32]. The mechanism by which depression is considered a risk factor is due to its association with producing atrophy in affected regions in AD. In addition, it can be related to neuropathological changes, so it could be potentially useful as a marker to identify MCI patients at risk of developing dementia [33]. 

The impact of these factors has been recently published by the Lancet Commission, which considers hypertension, obesity, hearing loss, head trauma, and excessive alcohol consumption in the middle stage of life as risk factors for cognitive impairment (45–65 years); and diabetes, smoking, depression, social isolation, diabetes mellitus, air pollution, and physical inactivity in the advanced stage of life (>65 years), for which it is proposed that by controlling each of these risk factors, the prevalence of dementia could be reduced, based on the fraction of risk attributable to the population [34].

Another important outcome in our study is the carrier of the ApoE ε4 allele status, which was not associated with the MCI condition, despite the evidence of its association.

As a risk factor for cognitive decline and neurodegeneration, there is still controversy regarding this association in the Mexican population [35]. These findings are in line with those of Villalpando et al., who reported a low prevalence of the ε4 allele in Mexican older adults. Furthermore, these authors did not demonstrate association between the ε4 allele and Alzheimer’s Disease (OR: 1.15, 95% CI: 0.54–2.47. *p* = 0.123) [36]. González et al., in a study that included 6377 Hispanic/Latinos, with a range age between 50 and 86 years, and MCI diagnosis did not demonstrate an association with both the ApoE ε4 or ApoE ε2 genotype, suggesting that the etiology of MCI could be different between Africans and Latinos, with an Amerindian mix vs. people with European ancestry, possibly as a result of colonizers in southern Europe, having a lower ApoE ε4 prevalence compared to northern Europeans. ApoE genotypes in Mexican-Americans, Mexican Mestizos and Mayans, showed a higher frequency of the ε3 allele (89–92%) and a lower one for the ε4 (6.9 to 8.4%), and ε2 (lower or absent) alleles [37].

This study has several limitations that must be recognized; cross-sectional design, and the small size and low statistical power of the sample to detect the influence of APOE ε4 as an association factor of aMCI and naMCI or to determine a dose-response effect between heterozygotes and homozygotes was not possible. In addition, we did not include other genetic polymorphisms that could be helpful in determining the effect of the genotype, and its respective alleles, over Mexican population with MCI. However, the main strength of our study is that currently there are not enough Latin American or Mexican studies that analyze the association between the ApoE genotype and cardiovascular risk factors and their potential interaction in two different forms of MCI.

## 5. Conclusions

Our study shows that cardiovascular risk factors represent the main associated factors for aMCI and naMCI after controlling for possible confounding factors. On the other hand, despite the fact that multiple studies in various ethnic groups (European) have shown that the APOE genotype is a risk factor for MCI, in the mestizo population the carrier status of the APOE ε4 allele did not contribute to this risk. However, more studies in the mestizo population are necessary to know the role of this or other polymorphisms and its relationship with MCI, since this could support the establishment of future strategies in the diagnosis and treatment of this disease in the mestizo population.

## Figures and Tables

**Table 1 brainsci-11-00068-t001:** Sociodemographic and health characteristics, cognitive performance scores, cardiovascular risk factors, and ApoE Genotype in the NC, aMCI and naMCI groups.

	Total (*n* = 137) Mean ± SD	NC (*n* = 63) Mean ± SD	aMCI (*n* = 24) Mean ± SD	naMCI (*n* = 50) Mean ± SD	*p* Value
Age, years ^a,b,c^	71.6 (7.23)	71.3 (6.2)	78.2 (7.4)	74.2 (7.2)	<0.001
Female	99 (72.3)	53 (84.1%)	15 (62.5%)	31 (62%)	0.017
Education, years ^a,c^	12.7 (5.04)	13.7 (3.7)	9.6 (5.9)	12.6 (5.6)	0.005
ABVD	5.8 (0.36)	5.8 (0.4)	5.9 (0.2)	5.9 (0.2)	0.150
AIVD	7.5 (1.06)	7.61 (1.0)	7.59 (1)	7.35 (1.0)	0.416
MMSE ^a,b^	27.8 (1.9)	28.6(1.4)	26.5 (2.1)	27.3 (1.9)	<0.001
MoCA ^a,b^	24.1 (3.6)	26.1 (2.5)	21.8 (3.3)	22.1 (3.7)	<0.001
NEUROPSI ^a,b,c^	101.5 (13.94)	110.8 (9.04)	85.2 (11.3)	98.1 (9.41)	<0.001
GDS ^a,c^	2.3 (2.2)	1.83(1.76)	3.6(2.42)	2.5 (2.53)	0.005
**Cardiovascular Risk Factors**					
Smoking status *n* (%)	59 (43%)	30 (48%)	7 (32%)	22 (45%)	0.433
Hypertension *n* (%)	78 (57%)	34 (54%)	13 (59%)	31 (66%)	0.449
Dyslipidemia *n* (%)	54 (39.4%)	19 (30%)	14 (64%)	21 (45%)	0.018
Diabetes *n* (%)	34 (25.8%)	12 (19%)	5 (23%)	17 (36%)	0.119
Obesity *n* (%)	36 (27.2%)	14 (22%)	5 (23%)	17 (35%)	0.299
Hypothyroidism *n* (%)	34 (24.8%)	11 (18%)	8 (36%)	15 (32%)	0.051
Stroke *n* (%)	10 (7.3%)	1 (2%)	5 (9%)	4 (23%)	<0.001
CHD *n* (%)	12 (8.7%)	4 (6%)	5 (6%)	3 (18%)	0.051
CVRF mean(SD) ^a,b^	1.9 (1.4)	1.5 (1.5)	2.5 (1.5)	2.2 (1.5)	0.004
**ApoE Genotype**					
ε2/ε3 *n* (%)		0 (0%)	0 (0%)	1 (2%)	
ε3/ε3 *n* (%)		48 (76%)	19 (79%)	30 (60%)	
ε3/ε4 *n* (%)		14 (22%)	4 (17%)	18 (36%)	
ε4/ε4 *n* (%)		1 (2%)	1 (4%)	1 (2%)	

NC, normal cognition; aMCI, amnestic mild cognitive impairment, naMCI, non-amnestic mild cognitive impairment, Katz, Index of Independence in Activities of Daily Living; Lawton, Index of Instrumental Activities of Daily Living; MMSE: Minimental Examination Test, MoCA: Montreal Cognitive Assessment; NEUROPSI, brief Neuropsychological Evaluation in Spanish; GDS, Geriatric Depression Scale; CHD, coronary heart disease; CVRF, cardiovascular disease risk factors. Post Hoc Analysis; significant difference < 0.05 was found between ^a^ the NC and aMCI group, ^b^ NC and naMCI group, ^c^ aMCI and naMCI groups.

**Table 2 brainsci-11-00068-t002:** Univariate multinomial logistic regression analysis of clinical and socio-demographic characteristics among ApoE ε4 carrier status on amnestic and non -amnestic versus NC (*n* = 39).

	aMCI		naMCI	
	Odds Ratio 95% (CI)	*p*	Odds Ratio 95% CI	*p*
Age	1.16 (1.07–1.25)	<0.001	1.06 (1.01–1.13)	0.022
Female	2.6 (.29–23.85)	0.380	3.60 (.76–17.00)	0.106
Education	0.82 (.72–0.92)	0.002	0.95 (0.88–1.04)	<0.001
GDS	1.52 (1.17–1.99)	0.002	1.16 (0.96–1.40)	0.119
CVRF	1.70 (1.18–2.46)	<0.001	1.48 (1.10–1.00)	0.007
**NEUROPSI:**				
Concentration and attention	0.70 (0.60–0.83)	<0.001	0.80 (0.71–0.90)	<0.001
Memory	0.67 (0.57–0.80)	<0.001	0.77 (0.69–0.86)	<0.001
Language	0.33 (0.20–0.55)	<0.001	0.75 (0.58–0.98)	0.035
Executives functions	0.76 (0.60–0.83)	<0.001	0.96 (0.71–0.90)	<0.001

CI: confidence interval, NC: normal cognition, aMCI: amnestic mild cognitive impairment. naMCI: non-amnestic mild cognitive impairment, GDS: Geriatric Depression Scale, CVRF: Cardiovascular Risk Factors, NEUROPSI: Brief Neuropsychological Evaluation in Spanish.

**Table 3 brainsci-11-00068-t003:** Multiple Regression effect of the ApoE ε4carrier status on the NC, amnestic and non-amnestic MCI groups.

**A. Unadjusted Model**						
	**aMCI**			**naMCI**		
	**Odds Ratio (95% CI) *p***			**Odss Ratio (95% CI) *p***		
ApoE ε4 carrier status	0.842 (0.26–2.64) *p* = 0.768		1.11 (0.12–10.08) *p* = 0.928	1.96 (0.86–4.25) *p* = 0.105		1.68 (0.35–7.36) *p* = 0.488
CVRF		1.70 (1.18–2.46) *p* = 0.004 *	1.7 (1.13–2.65) *p* = 0.011 *		1.48 (1.10–1.99) *p* = 0.009*	1.43 (1.00–2.09) *p* = 0.049 *
ApoE ε4 carrier status and CVRF			0.95 (0.40–2.27) *p* = 0.904			0.78 (0.57–2.13) *p* = 0.776
**B. Adjusted Model**						
	**aMCI**			**naMCI**		
	**Odds Ratio (95% CI) *p***			**Odds Ratio (95% CI) *p***		
ApoE ε4 carrier status	1.06 (0.268–2.641) *p* = 0.936		1.78 (0.15–21.94) *p* = 0.648	1.93 (0.80–4.65) *p* = 0.143		1.50 (0.33–6.91) *p* = 0.602
CVRF		1.51 (1.01–2.27) *p* = 0.018 *	1.62 (0.99–2.64) *p* = 0.055 *		1.34 (0.97–1.92) *p* = 0.059*	1.28 (0.89–1.85) *p* = 0.178
ApoE ε4 carrier status and CVRF			0.82 (0.32–2.11) *p* = 0.675			1.13 (0.57–2.22) *p* = 0.725

NC, normal cognition; aMCI, amnestic mild cognitive impairment, naMCI, non-amnestic mild cognitive impairment. CVRF: Cardiovascular Risk Factors. Adjusted Model for age, educational level and sex. ApoE ε3/ ε4 and ε4/ε4 and carrier status*CVRF: interaction. CI: confidence interval. * *p* ≤ 0.05.

## Data Availability

No new data were created or analyzed in this study. Data sharing is not applicable to this article.

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
