# Peer review of "Association between ApoE ε4 Carrier Status and Cardiovascular Risk Factors on Mild Cognitive Impairment among Mexican Older Adults"

_brainsci, 2021, doi:10.3390/brainsci11010068_

Round 1

Reviewer 1 Report

Overall: APOE 4 has been a well-established risk factor for MCI and AD. How do the authors reconcile their negative results with the literature?

Introduction: The language could use editing, especially in terms of academic style of writing and the fluency of sentences, examples: line 53 “seems not to be universal” could be phrased as “ does not seem to be universal”.

Methods: For the diagnosis of MCI, is the diagnosis corroborated by imaging, such as the MRI data?

Limited by sample size, the APOE4 status was binomially classified. However, readers would wonder had the APOE status also be classified non-binomially (by the number of alleles), how would the analyses and findings turn out to be? Upon seeing Table 1 with only one subject with double e4 alleles, this should be cited as a limitation and one-liner explanation.

CVRF: 1) should be moved to exposure/ moderating variable, since it is not just a covariate.

2) is there specific references to how and why these eight conditions were selected?

Statistical analysis:

Is the study powered to detect significant association (or lack thereof) between APOE 4 alleles with cognitive status? This point has to be clear.

chi-square to compare qualitative data sounds wrong- chi square can’t analyse qualitative data.

From the descriptions, there seemed that many statistical tests were run, and this could present the issue of false positives. I suggest to address this potential issue in the last sentence(s) of the section.

Table 2 is confusing. Readers won’t know what are the exposure and outcome measures by reading only the table. The alignments also need to be improved.

Table 3. Suggest that “APOE e4 carrier status” to be labelled fully in the table, else it risks misleading the readers. The table could be split into a & b to clearly demarcate the adjusted and unadjusted models.

Discussion:

This section is way too long. Suggest to limit to at most 7 paragraphs.

Line 193 “evolution” should be changed to “trajectory”. MCI could not evolve but can only progress/ disgress.

My main concern is that the authors claimed that compared to the CVS factors, APOE genotype does not contribute to the risk of MCI. I would be very careful not to arrive at such as conclusion, especially as the sample size is relatively small and most likely underpowered to claim this. Related to this, the conclusion could be phrased in a better light to summarize the finding on significant CVS risk factors.

Line 197-198: Another main concern is the novelty and main contribution of this study to the literature, esp. considering the overwhelming amount of literature in the same topic? Perhaps one novelty is the sample (ethnicity). However, many studies of much larger sample sizes have been conducted in Mexican before, as cited. The distinction between what this study contributed differently needs to be clearly delineated. Another novelty could perhaps be the interaction effect.

The third main concern is that the authors highlighted progression of MCI to dementia many times throughout the paper. However, is progression to dementia longitudinally examined? If not, this claim should be toned down, as it is mostly unrelated to the central themes of the paper.

Author Response

Reviewer 1:

1.- Overall: APOE 4 has been a well-established risk factor for MCI and AD. How do the authors reconcile their negative results with the literature?

Response: I agree with your comment. This will be recognized at the limits. However, the low frequency of this genotype is supported by the findings in races other than the Caucasian. For example the reference that cites González et al., That included 6,377 Hispanic / Latinos, with a range age between 50 and 86 years, and MCI diagnosis did not demonstrate an association with both the ApoE ε4 or ApoE ε2 genotype, suggesting that the etiology of MCI could be different between Africans and Latinos, with an Amerindian mix vs people with European ancestry, possibly as a result of colonizers in southern Europe, having a lower ApoE ε4 prevalence compared to northern Europeans. ApoE genotypes in Mexican Americans, Mexican Mestizos and Mayans, showed a higher frequency of the ε3 allele (89-92%) and a lower one for the ε4 (6.9 to 8.4%), and ε2 (lower or absent) alleles.

2.- Introduction: The language could use editing, especially in terms of academic style of writing and the fluency of sentences, examples: line 53 “seems not to be universal” could be phrased as “ does not seem to be universal”.

Response: Yes of course, the change was done. Thanks for you observation, we have reviewed the edition and style of the manuscript and noted pertinent changes. 

3.- Methods: For the diagnosis of MCI, is the diagnosis corroborated by imaging, such as the MRI data?

Response: The MRI results were used by the researchers to rule out structural pathology that excluded the patient from the study, as to support the clinical diagnosis of the subtype of MCI. Thanks.

Page: 2 line: 86,87.

4.- Limited by sample size, the APOE4 status was binomially classified. However, readers would wonder had the APOE status also be classified non-binomially (by the number of alleles), how would the analyses and findings turn out to be? Upon seeing Table 1 with only one subject with double e4 alleles, this should be cited as a limitation and one-liner explanation.

Response: I agree with the observation and this has been highlighted as a limit attributed to the size of the sample, but it was decided to group in this way supported by the fact that the ε3 / ε4 heterozygous carrier state also represents a risk for the MCI condition.

Page: 2 Line: 51

Page: 8 Line: 286-289

5.- CVRF: 1) should be moved to exposure/ moderating variable, since it is not just a covariate.

Response: Excellent observation CVRFs are grouped as exposure variable. Thank you.

                  2) is there specific references to how and why these eight conditions were selected?

Response: We included the following variates exposure based on previous studies that have addressed cardiovascular risk  factors on Cognitive Impartment

6.- Statistical analysis:

a.- Is the study powered to detect significant association (or lack thereof) between APOE 4 alleles with cognitive status? This point has to be clear.

b.- chi-square to compare qualitative data sounds wrong- chi square can’t analyse qualitative data.

Response: chi square to compare categorical data, the change was done. Thanks. Page: 3 Line: 136

c.- From the descriptions, there seemed that many statistical tests were run, and this could present the issue of false positives. I suggest to address this potential issue in the last sentence(s) of the section.

Response: Thanks for your suggestion, the description of the statistical analysis is abbreviated

7.- Results:

a.- Table 2 is confusing. Readers won’t know what are the exposure and outcome measures by reading only the table. The alignments also need to be improved.

Response: We agree to improve the title and presentation of the data of the table 2.  The adjustments  are presented on page:  5 line: 172.

b.- Table 3. Suggest that “APOE e4 carrier status” to be labelled fully in the table, else it risks misleading the readers. The table could be split into a & b to clearly demarcate the adjusted and unadjusted models.

Response: According to the importance of the table being self-explanatory, the presentation of the data is modified to illustrate both models.

8.- Discussion:This section is way too long. Suggest to limit to at most 7 paragraphs.

Response: Thanks for the suggestion we review and edit the body of the discussion.

a.- Line 193 “evolution” should be changed to “trajectory”. MCI could not evolve but can only progress/ disgress.

Response: We agree with the observation the change was done. Page: 6. Line: 195

b.- My main concern is that the authors claimed that compared to the CVS factors, APOE genotype does not contribute to the risk of MCI. I would be very careful not to arrive at such as conclusion, especially as the sample size is relatively small and most likely underpowered to claim this. Related to this, the conclusion could be phrased in a better light to summarize the finding on significant CVS risk factors.

Response: Thanks for the observation de change was done.

c.- Line 197-198: Another main concern is the novelty and main contribution of this study to the literature, esp. considering the overwhelming amount of literature in the same topic? Perhaps one novelty is the sample (ethnicity). However, many studies of much larger sample sizes have been conducted in Mexican before, as cited. The distinction between what this study contributed differently needs to be clearly delineated. Another novelty could perhaps be the interaction effect.

Response: Thanks to your recommendation, we have made modifications to the discussion and conclusions to improve their meaning and to be able to transmit our results more objectively.

d.- The third main concern is that the authors highlighted progression of MCI to dementia many times throughout the paper. However, is progression to dementia longitudinally examined? If not, this claim should be toned down, as it is mostly unrelated to the central themes of the paper.

Response: Thank you for your recommendation, we have made changes to focus attention on MCI and not on dementia.

REVIEWER  2

1.-  ApoE status is the most accepted risk factor for Alzheimer’s in the AD community. In this study there was only 1 individual with an ApoE4 homozygous status in each group, so as admitted by the author, this study is way underpowered to discount ApoE status as an important factor. It is not the only factor but it is an accepted and heavily researched and scientifically supported risk factor for dementia.

Response: I agree with your comment. This will be recognized at the limits. However, the low frequency of this genotype is supported by the findings in races other than the Caucasian. Although lack of power is recognized as such, this finding is not the only one that has been described in a non-Caucasian population.

2.- Rewrite sentences 42-43 making it more clear that it is MCI you would like them to focus on and not dementia itself which is how it currently reads. 

Response:  We agree.  The change was done

Page: 1 Line: 38,39.

3.- 44-45 not all MCI presents exactly the same so soften the generalities. “a slight damage” for example is very vague and not a very scientific/clinically relevant way to depict pathology and not always the case

Response: According to the observation the term "slight" is eliminated.

4.- 52 insert for before both. Also, there are risks when you are heterozygous for e3 so these need to be shared. e2 and e3 are not outright protective. Comparisons between homozygous and heterozygous ApoE types are overlooked.

Response: Accordingly, we have included the homozygous and heterozygous condition at the beginning of the paragraph to clarify the carrier status

Page: 2, Line: 49.

5.- 69 CVRFR instead of CVRF

Response: The change was done. Thanks.

6.- 82 is “illiterates” replaceable as it is a very derogatory word

Response: without schooling

7.- 86-87 “as to support the clinical diagnosis” This sentence is confusing

Response: The sentence was changed. Thanks.

8.- 93-97 “in diagnosing that conditions” “was administered”. Restructure this sentence

Response: The change was done. Thank a lot.

Page:  2, 3: Line: 93-96.

9.--99 “Administration time range…” not sure what this is referring to. Restructure

Response: The sentence was restructured. Page. 3 Line: 98  

10.- 117 space before “for women”

Response: The change was done.

11.- 113 “NC controls” NC or spell out normal controls in full

Response: The change was done

12.- 119-120 What do 1 and I stand for in the primer design?

Response: The sentence was restructured. Eliminated: First and “primer”

Page: 3 Line: 117,118.

13.- 123-124 “then loaded with polyacrylamide” rephrase

Response: The change was done.

Page:  3 Line: 123

14.- 142 “in order of testing” rephrase

Response: The sentence was restructured.

Line: 146,147.

15.- 143 “confidential” instead of confidence

Response: The change was done.

Pag: 4 Line: 148

16.- 158 spacing

Response: The change was done

17.- 206-207 sentence needs restructuring

Response: The change was done. Page 6. Line: 210.

18.- 236-237 “microateromas and lipohyalinosis”?

Response: the change was done .

19.- 256 “super-added” rephrase

Response: Super-added was removed.

Page: 8 line: 285 

20.- 286 “several limits” instead of several limitations

Response: The change was done.

21.- 286-289 stating that this study is too low in power to make the statements that it shared is problematic.

Response: Thank you for your recommendation, we have included in limits the low power of the study and detailed in the conclusions the strengths of the same.

Reviewer 2 Report

ApoE status is the most accepted risk factor for Alzheimer’s in the AD community. In this study there was only 1 individual with an ApoE4 homozygous status in each group, so as admitted by the author, this study is way underpowered to discount ApoE status as an important factor. It is not the only factor but it is an accepted and heavily researched and scientifically supported risk factor for dementia.

Other points:

Rewrite sentences 42-43 making it more clear that it is MCI you would like them to focus on and not dementia itself which is how it currently reads. 

44-45 not all MCI presents exactly the same so soften the generalities. “a slight damage” for example is very vague and not a very scientific/clinically relevant way to depict pathology and not always the case

52 insert for before the both. Also there are risks when you are heterozygous for e3 so these need to be shared. e2 and e3 are not outright protective. Comparisons between homozygous and heterozygous ApoE types are overlooked.

69 CVRFR instead of CVRF

82 is “illiterates” replaceable as it is a very derogatory word

86-87 “as to support the clinical diagnosis” This sentence is confusing

93-97 “in diagnosing that conditions” “was administered”. Restructure this sentence

99 “Administration time range…” not sure what this is referring to. Restructure

117 space before “for women”

113 “NC controls” NC or spell out normal controls in full

119-120 What do 1 and I stand for in the primer design?

123-124 “then loaded with polyacrylamide” rephrase

142 “in order of testing” rephrase

143 “confidential” instead of confidence

158 spacing

206-207 sentence needs restructuring

236-237 “microateromas and lipohyalinosis”?

256 “super-added” rephrase

286 “several limits” instead of several limitations

286-289 stating that this study is too low in power to make the statements that it shared is problematic.

Author Response

(The authors gave the same response as above.)

Round 2

Reviewer 1 Report

  1. 2) is there specific references to how and why these eight conditions were selected?

Response: We included the following variates exposure based on previous studies that have addressed cardiovascular risk  factors on Cognitive Impartment

Reviewer: they are indeed the established risk factors for cognitive impairment. However, what is needed is to cite the relevant literature, especially in reference to the construct of the CVRF. Since as it is now, it seems that the CVRF is a construct that the authors come out with themselves. However, it is unclear if it is so. If it was a novel construct, then further validation is needed.

6.- Statistical analysis:

a.- Is the study powered to detect significant association (or lack thereof) between APOE 4 alleles with cognitive status? This point has to be clear.

Reviewer: Lacking an author response here. I appreciate that the authors have made amendments to the limitation, partially addressing this critical limitation that severely limits the robustness and validity of the findings and conclusions. In view of this, the limitation should be expanded to include an explanation that the lack thereof of the association between the APOE 4 alleles with cognitive status could well be due to the small sample size. This would be an important information for the field and the readers to understand.

c.- From the descriptions, there seemed that many statistical tests were run, and this could present the issue of false positives. I suggest to address this potential issue in the last sentence(s) of the section.

Response: Thanks for your suggestion, the description of the statistical analysis is abbreviated

Reviewer: The authors have misunderstood the point made here. What the reviewer would like to point out here is that since multiple ANOVA and regressions were performed, there are likely chances where a significant p-value could be due to chance alone. This presents the issues of multiple comparisons and needs to be addressed in the limitation section.

b.- Table 3. Suggest that “APOE e4 carrier status” to be labelled fully in the table, else it risks misleading the readers. The table could be split into a & b to clearly demarcate the adjusted and unadjusted models.

Response: According to the importance of the table being self-explanatory, the presentation of the data is modified to illustrate both models.

Reviewer: Thank you for addressing this point. “APOE e4 carrier status” should be explained in terms of their binomial (instead of heterogenous nature), in the footnote of the table.

8.- Discussion:This section is way too long. Suggest to limit to at most 7 paragraphs.

Response: Thanks for the suggestion we review and edit the body of the discussion.

Reviewer: A number of paragraphs in the discussion section need to be combined with other paragraphs. For example, lines 211-213 to be combined with previous paragraph to ensure a better flow.

c.- Line 197-198: Another main concern is the novelty and main contribution of this study to the literature, esp. considering the overwhelming amount of literature in the same topic? Perhaps one novelty is the sample (ethnicity). However, many studies of much larger sample sizes have been conducted in Mexican before, as cited. The distinction between what this study contributed differently needs to be clearly delineated. Another novelty could perhaps be the interaction effect.

Response: Thanks to your recommendation, we have made modifications to the discussion and conclusions to improve their meaning and to be able to transmit our results more objectively.

Reviewer: The result and conclusion sections have been improved. However, the distinction between what this study contributed differently needs to be clearly delineated and is still not clear in this revised version.

Author Response

Reviewer:  is there specific references to how and why these eight conditions were selected?

RESPONSE ROUND 2

No. The eight selected conditions are widely known risk factors for the development of cognitive impairment. However, in the present study CVRF variable simply represent a sum of them where higher number represents more cardiovascular risk factors.

Reviewer: they are indeed the established risk factors for cognitive impairment. However, what is needed is to cite the relevant literature, especially in reference to the construct of the CVRF. Since as it is now, it seems that the CVRF is a construct that the authors come out with themselves. However, it is unclear if it is so. If it was a novel construct, then further validation is needed.

RESPONSE ROUND 2:

Thanks for the comment. Indeed, as the reviewer points out, the eight conditions are well-established risk factors for cognitive impairment in the literature. The CVRF variable represents only the sum of them under the hypothesis that having two will be worse than only having one or none, and so on. It is not about a new construct. This will be specified in the methods section to avoid confusion for the reader.

6- Statistical analysis:

a.- Is the study powered to detect significant association (or lack thereof) between APOE 4 alleles with cognitive status? This point has to be clear.

Reviewer: Lacking an author response here. I appreciate that the authors have made amendments to the limitation, partially addressing this critical limitation that severely limits the robustness and validity of the findings and conclusions. In view of this, the limitation should be expanded to include an explanation that the lack thereof of the association between the APOE 4 alleles with cognitive status could well be due to the small sample size. This would be an important information for the field and the readers to understand.

RESPONSE ROUND 2

Thanks for concern. Indeed, our study is a secondary analysis whose power was not estimated precisely for the present study. This will be recognized both in the methods as well as in the limits of the study. Line 141-144

c.- From the descriptions, there seemed that many statistical tests were run, and this could present the issue of false positives. I suggest to address this potential issue in the last sentence(s) of the section.

RESPONSE: Thanks for your suggestion, the description of the statistical analysis will be abbreviated

Reviewer: The authors have misunderstood the point made here. What the reviewer would like to point out here is that since multiple ANOVA and regressions were performed, there are likely chances where a significant p-value could be due to chance alone. This presents the issues of multiple comparisons and needs to be addressed in the limitation section.

RESPONSE ROUND 2  

Response:  Thanks for the clarification. This problem will be recognized in the manuscript as suggested by the reviewer.Since many tests were run, it cannot be ruled out that some statistically significant results are due to chance alone, for which careful interpretation is required.

b.- Table 3. Suggest that “APOE e4 carrier status” to be labelled fully in the table, else it risks misleading the readers. The table could be split into a & b to clearly demarcate the adjusted and unadjusted models.

Response: According to the importance of the table must be self-explanatory, data presentation was modified to illustrate both models.

Reviewer: Thank you for addressing this point. “APOE e4 carrier status” should be explained in terms of their binomial (instead of heterogenous nature), in the footnote of the table.

RESPONSE ROUND 2

Response: the table was fixed. Line 201-202

8.- Discussion:This section is way too long. Suggest to limit to at most 7 paragraphs.

Response: Thanks for the suggestion. We review and edit the body of the discussion.

Reviewer: A number of paragraphs in the discussion section need to be combined with other paragraphs. For example, lines 211-213 to be combined with previous paragraph to ensure a better flow.

RESPONSE ROUND 2

Response: the discussion was fixed.

c.- Line 197-198: Another main concern is the novelty and main contribution of this study to the literature, esp. considering the overwhelming amount of literature in the same topic? Perhaps one novelty is the sample (ethnicity). However, many studies of much larger sample sizes have been conducted in Mexican before, as cited. The distinction between what this study contributed differently needs to be clearly delineated. Another novelty could perhaps be the interaction effect.

Response: Thanks for your recommendation. We have made modifications to the discussion underlining our contributions as well as conclusions were improved in order  to be able to transmit our results more objectively.

Reviewer: The result and conclusion sections have been improved. However, the distinction between what this study contributed differently needs to be clearly delineated and is still not clear in this revised version.

RESPONSE ROUND 2

As mentioned above, a further review of the discussion and conclusions was made outlining our contributions. Line: 292-302.